# Inflammation and Tumor Progression: The Differential Impact of SAA in Breast Cancer Models

**DOI:** 10.3390/biology13090654

**Published:** 2024-08-23

**Authors:** Daniel Wilhelm Olivier, Carla Eksteen, Manisha du Plessis, Louis de Jager, Lize Engelbrecht, Nathaniel Wade McGregor, Preetha Shridas, Frederick C. de Beer, Willem J. S. de Villiers, Etheresia Pretorius, Anna-Mart Engelbrecht

**Affiliations:** 1Department of Physiological Sciences, Faculty of Science, Stellenbosch University, Stellenbosch 7600, Western Cape, South Africawimdv@sun.ac.za (W.J.S.d.V.);; 2Division of Anatomical Pathology, Department of Pathology, Faculty of Medicine and Health Sciences, Stellenbosch University, Stellenbosch 7600, Western Cape, South Africa; 3National Health Laboratory Service, Tygerberg Hospital, Cape Town 7505, Western Cape, South Africa; 4Central Analytical Facilities, Fluorescence Microscopy Unit, Stellenbosch University, Stellenbosch 7600, Western Cape, South Africa; 5Department of Genetics, Faculty of Agrisciences, Stellenbosch University, Stellenbosch 7600, Western Cape, South Africa; nwm@sun.ac.za; 6Department of Psychiatry, Faculty of Medicine and Health Sciences, Stellenbosch University, Cape Town 7505, Western Cape, South Africa; 7Department of Internal Medicine, University of Kentucky, Lexington, KY 40536, USA; 8Department of Internal Medicine, Faculty of Medicine and Health Sciences, Stellenbosch University, Cape Town 7505, Western Cape, South Africa; 9Department of Global Health, African Cancer Institute (ACI), Faculty of Medicine and Health Sciences, Stellenbosch University, Stellenbosch 7505, Western Cape, South Africa

**Keywords:** breast cancer, inflammasome, metastasis, serum amyloid A

## Abstract

**Simple Summary:**

Previous research highlights the involvement of the Serum Amyloid A (SAA) protein family in inflammation and various diseases. This study explores the role of SAA1 and SAA2 (SAA1/2) in triple-negative breast cancer using a mouse model. Syngeneic breast tumors were established in wild-type and SAA1/2-deficient (SAADKO) mice, with assessments of tumor volume, survival, inflammatory profiles, and tumor characteristics.

**Abstract:**

**Background:** Previous research has shown that the Serum Amyloid A (SAA) protein family is intricately involved in inflammatory signaling and various disease pathologies. We have previously demonstrated that SAA is associated with increased colitis disease severity and the promotion of tumorigenesis. However, the specific role of SAA proteins in breast cancer pathology remains unclear. Therefore, we investigated the role of systemic SAA1 and SAA2 (SAA1/2) in a triple-negative breast cancer mouse model. **Methods:** Syngeneic breast tumors were established in wild-type mice, and mice lacking the SAA1/2 (SAADKO). Subsequently, tumor volume was monitored, species survival determined, the inflammatory profiles of mice assessed with a multiplex assay, and tumor molecular biology and histology characterized with Western blotting and H&E histological staining. **Results:** WT tumor-bearing mice had increased levels of plasma SAA compared to wild-type control mice, while SAADKO control and tumor-bearing mice presented with lower levels of SAA in their plasma. SAADKO tumor-bearing mice also displayed significantly lower concentrations of systemic inflammatory markers. Tumors from SAADKO mice overall had lower levels of SAA compared to tumors from wild-type mice, decreased apoptosis and inflammasome signaling, and little to no tumor necrosis. **Conclusions:** We demonstrated that systemic SAA1/2 stimulates the activation of the NLRP3 inflammasome in breast tumors, leading to the production of pro-inflammatory cytokines. This, in turn, promoted apoptosis and tumor necrosis but did not significantly impact tumor growth or histological grading.

## 1. Introduction

Inflammation plays a key role in carcinogenesis through chronic cytokine/chemokine signaling, aiding tumor growth, suppressing regulated cell death programs such as apoptosis, and promoting metastasis [e.g., epithelial to mesenchymal transition (EMT)] by changing the tumor microenvironment (TME) [1]. Central to this signaling framework is the family of Serum Amyloid A (SAA) proteins, induced by various cytokines such as interleukin (IL)-1β, IL-6, and tumor necrosis factor alpha (TNFα) in response to local inflammation or tissue injury [2].

Significant evidence has shown the almost omnipresent nature of Serum Amyloid A (SAA) in the blood of cancer patients [3,4,5,6]. The most notable discovery was that SAA is upregulated in many cell types, including macrophages, adipocytes, and fibroblasts, which are commonly infiltrated within tumors [7], in addition to cancer cells themselves [8,9,10,11]. To date, SAA has been associated with a higher tumor stage, the presence of lymphovascular invasion and lymph node metastasis, increased hypoxia-inducible factor 1 alpha (*HIF1α*) expression, and the production of many pro-inflammatory cytokines [12]. Our research group previously reported similar findings in a mouse model of colitis-associated cancer [13]. SAA, therefore, has a central role during inflammation and is induced as part of the inflammatory response to control subsequent cytokine signaling. However, the exact functional role of elevated SAA levels and its presence in breast tumor tissue has largely remained unclear.

Another important mediator of inflammation, which has also emerged as a mediator of cancer progression, is the inflammasome. Inflammasomes are multi-protein complexes, composed of pro-caspase-1, a NOD-like receptor, and an apoptosis-associated speck-like protein containing a caspase activation and recruitment domain (CARD), which, upon pathogen- or damage-associated stimuli, are assembled. This promotes the infiltration of inflammatory and immunocompetent cells into the inflamed site during the innate immune response. Furthermore, Nod-like receptor (NLR) proteins can also be activated by the binding of SAA to toll-like receptors (TLRs), which have also emerged as important mediators of cancer progression for their role in inflammation [14]. The best characterized is the NLR family pyrin domain containing 3 (NLRP3) inflammasome [15]. Briefly, TLRs are autophosphorylated in response to pathogen-associated molecular patterns/damage-associated molecular patterns (PAMPs/DAMPs), leading to nuclear factor kappa beta (NFκB) activation, transcription of pro-IL-1β, and pro-IL-18, cytosolic assembly of multimeric NLRP3 inflammasome complexes, caspase 1 activation, and finally IL-1β and IL-18 maturation [16]. Although it has been reported that several types of amyloids can activate the NLRP3 inflammasome, including SAA and amyloid-β, the role of SAA signaling in NLRP3 activation in breast cancer is limited. The aim of this study was therefore to investigate the effects of SAA1/2 in an in vivo triple negative breast cancer model (TNBC), in mice lacking both *SAA1* and *SAA2*. Here, we show the novel role of SAA1/2 in modulating the NLRP3 inflammasome, apoptosis, and necrosis in breast cancer.

## 2. Materials and Methods

### 2.1. Animals, Housing, and Genotyping

#### 2.1.1. Animals and Housing

Ethical clearance was obtained from the Stellenbosch University Research Ethics Committee (ACU-2019-6426) and all procedures were performed under the committee’s guidelines. SAADKO breeding pairs were generously donated by Professors Frederick C de Beer and Maria C de Beer (University of Kentucky, Lexington, KY, USA) and bred alongside WT mice at Stellenbosch University at the Tygerberg Animal Facility.

Mice were housed under temperature-controlled conditions in a 12 h reversed dark–light cycle. Distilled water and a standard chow were available to all mice ad libitum. A total of 20 female WT and 20 female SAADKO mice were used in the study, each divided into two groups—control and EO771 tumor-induced mice.

#### 2.1.2. SAADKO Genotyping

Following weaning, tail clippings from each SAADKO mouse were sent for genotyping to the Centre for Proteomic and Genomic Research. Appendix A shows the report for the amplification of a single target fragment of ~303 bp, unique to the SAADKO mice [17].

### 2.2. EO771 SAA1/2/3/4 mRNA Screening, Tumor Induction, Monitoring, and Assessment

#### 2.2.1. EO771 SAA1/2/3/4 mRNA Screening

The C57BL/6 syngeneic murine breast cancer cell line, EO771, used for tumor induction, was screened for SAA1/2/3/4 mRNA expression. Herein, EO771 cells were lysed, and the sample enriched for mRNA, followed by cDNA synthesis and quality control, before PCR amplification of SAA1/2/3/4. Subsequently, a post-PCR clean-up was performed, and the resultant amplicons were sequenced with direct-cycle Sanger sequencing. The Appendix A describes the process in detail (Appendix A), lists the forward and reverse primers used for PCR amplification and Sanger sequencing (Appendix A), and shows the products of PCR amplification (Appendix A) and the Sanger sequencing results confirming SAA1/2/3/4 mRNA expression (Appendix A).

#### 2.2.2. Cell Culture

The C57BL/6 mycoplasma-free syngeneic murine breast cancer cell line, EO771, was generously provided by Fengzhi Li (Roswell Park Cancer Institute, Buffalo, NY, USA). EO771 cells were cultured in Dulbecco’s Modified Eagles Medium (DMEM) (Gibco^®^, Thermo Fisher Scientific, Johannesburg, South Africa), supplemented with 10% fetal bovine serum (FBS) (Capricorn Scientific, Whitehead Scientific, Cape Town, South Africa) and 1% Penicillin-Streptomycin solution (Gibco^®^, Thermo Fisher Scientific, Johannesburg, South Africa). Cells were passaged using the TrypLE^™^ Express enzyme (Gibco^®^, Thermo Fisher Scientific, Johannesburg, South Africa) and maintained in a humidified incubator at 5% CO_2_ and 37 °C. Before injection, cells were detached using TrypLE^™^ Express, cells were pelleted and were then washed 3X with Hank’s Balanced Salt Solution (HBSS) (Merck, Johannesburg, South Africa) before resuspension in HBSS. Subsequently, cells were counted using a Countess^™^ Automated cell counter and Countess^™^ cell counting kit, before being diluted to 1.5 × 10^6^ cells/100 μL. Finally, 100 µL aliquots were aliquoted into syringes fitted with sterile 23-gauge needles, ready for injection.

#### 2.2.3. Tumor Establishment

EO771 cells were subcutaneously injected into 9-week-old WT and SAADKO mice under 3% Isoflurane (Isofor, Safeline Pharmaceuticals, Johannesburg, South Africa), sedation at the fourth mammary fat pad of each mouse. Control mice were injected with 100 μL HBSS solution at the same site and conditions.

#### 2.2.4. Assessment and Measurements

Following injections, mice were closely monitored for 3 days to ensure no physical deterioration, whereafter the mice were weighed and assessed every second day. After the first observation of a palpable tumor in tumor-bearing mice, tumors were measured every second day with a digital caliper, and tumor volumes were calculated as follows:$$Tumor\;volume\;({mm}^{3})=\frac{W^{2}\times L}{2}$$

W represents the width and L the length of the tumor. However, to ensure mice with larger tumors do not deteriorate unnoticed, mice bearing tumors larger than 200 mm^3^ were assessed daily. Humane endpoints were set at a weight loss of larger than 10% of the previous measurement; discovery of bleeding ulcers; or sudden physical deterioration based on the mouse grimace scale.

### 2.3. Euthanasia and Sample Collection

Upon reaching the experimental (300–400 mm^3^) or humane endpoint, mice were euthanized under 3% Isoflurane sedation. Blood samples were collected through cardiac puncture with a 23-gauge needle and death was confirmed by cervical dislocation. Before cervical dislocation, each mouse was exsanguinated, and the blood was collected into citrate pediatric tubes (PathCare, Cape Town, South Africa). Plasma was collected from blood samples through centrifugation at 1000× *g* (10 min) using a Labnet Prism Microcentrifuge, aliquoted into 50 μL aliquots, and stored at −80 °C until the day before analysis. Tumor samples were weighed and divided into two sections. One section per sample was snap-frozen, using liquid nitrogen, and stored at −80 °C for further analysis with Western blotting. The remaining sections were stored in 4% neutral buffered formaldehyde (NBF) for tumor histopathology.

### 2.4. Tumor Growth and Species Survival

#### Tumor Growth Analysis

Tumor growth for each species was analyzed using a mixed model analysis of variance (ANOVA) in R (lmer package, Kenward–Roger’s degrees of freedom; R Core Team, version 4.4.1). Subsequently, tumor growth for each species was plotted as the least-squares mean ± standard error of the mean (SEM). Mice that reached the experimental endpoint, or in the unfortunate event of a humane endpoint, were sacrificed, and the date was recorded for survival analysis. Subsequently, the data were statistically analyzed using a Cox–Mantel Test, and the data plotted using Kaplan–Meier cumulative proportional survival. All humane endpoints were censored.

### 2.5. Blood Plasma Inflammatory Profiling

#### 2.5.1. Serum Amyloid A

Plasma SAA levels for control and tumor-bearing mice were assessed with Western blot, using rabbit anti-mouse SAA antisera (De Beer laboratory, University of Kentucky, USA). In short, total protein was determined by the Bradford method, whereafter 50 µg protein was prepared in 2X Laemli’s sample buffer at a ratio of 2:1 before SDS-PAGE and Western blot analysis were performed.

#### 2.5.2. IL-1β, IL-6, IL-10, MCP-1, and TNFα

The inflammatory profile of tumor-bearing mice was assessed for IL-1β, IL-6, IL-10, MCP-1, and TNFα. Briefly, plasma samples were transferred to −20 °C on the day before analysis. On the day of analysis, samples were thawed on ice and 50 μL of each sample was diluted 1:2 with the kit constituents. Subsequently, diluted samples were subjected to a Milliplex MAP Mouse cytokine/chemokine magnetic bead panel (MCYTOMAG- 70K, Merck, Johannesburg, South Africa) according to the manufacturer’s protocol and detected by Luminex xMAP (Luminex Corporation).

### 2.6. SDS-PAGE and Western Blot Analysis of Tumor Tissue

#### 2.6.1. Protein Harvest from Tumor Tissue

Tumor tissue was cut into smaller pieces and homogenized in ice-cold radioimmunoprecipitation assay (RIPA) buffer [0.1% Sodium dodecyl sulfate (SDS), 5 mM Ethylene glycol-bis (2-aminoethyl ether)- tetraacetic acid, 5 mM Ethylenediaminetetraacetic acid, 1% Sodium deoxycholate, 1% NP-40, 154 mM Sodium Chloride (NaCl), 65 mM Tris-base, pH 7.4], containing the protease and phosphatase inhibitors Leupeptin (1 μg/mL), Aprotinin (1 μg/mL), Benzamide (1 μg/mL), Sodium orthovanadate (1 mM), Sodium Fluoride (1 mM), and phenylmethane sulfonyl fluoride (1 mM). Subsequently, homogenates were incubated on ice for 2 h, before being centrifuged for 20 min at 4 °C and 10,000× *g*, using a Labnet Spectrafuge 16M. Finally, the supernatants were transferred to new pre-cooled Eppendorf tubes, before being stored at −80 °C.

#### 2.6.2. Protein Quantification and Sample Preparation

Before protein quantification, samples were thawed on ice. Total protein was determined using the Bradford method [18]. Samples were diluted 40× before the addition of Bradford reagent and spectrophotometric measurement at 595 nm on a Cecil Aurius CE 2021 spectrophotometer.

Samples (50 µg) were prepared in Laemli’s sample buffer. Samples were then pulse centrifuged for 10 s and boiled for 5 min at 95 °C on a heating block.

#### 2.6.3. SDS-PAGE and Western Blot

Samples and BLUeye Prestained Protein ladders (Sigma Aldrich) were loaded and separated on TGX FastCast 12% Acrylamide gels (BioRad). Electrophoreses were carried out at 100 V for approximately 90 min. Gels were exposed for 2.5 min to UV light to activate the Stain- Free^™^ properties, using the ChemiDoc^™^ MP System (BioRad, Lasec, Johannesburg, South Africa).

Proteins were subsequently transferred to a polyvinylidene fluoride (PVDF) membrane using the Trans-Blot^®^ Turbo^™^ RTA Midi PVDF Transfer kit (BioRad, Lasec, Johannesburg, South Africa), per the manufacturer’s instructions, and a Trans-Blot^®^ Turbo Transfer System (BioRad). The successful transfer was visualized on a ChemiDoc^™^ MP System, followed by a blocking step in 5% fat-free milk for 120 min. Thereafter, membranes were washed three times in TBS-T for 5 min, before being probed with the appropriate primary antibody overnight at 4 °C.

The following day, membranes were washed (5X, 5 min, TBS-T) and probed with the appropriate secondary antibody at room temperature for 60 min, followed by another wash step (5X, 5 min, TBS-T). Finally, the membranes were developed using the Clarity^™^ ECL Substrate (BioRad) on the ChemiDoc™ MP System. Analyses were performed with Image Lab Software (Bio-Rad, version 6.1, Hercules, CA, USA), using the total protein content of each membrane and a standard protein sample for normalization. Antibody suppliers, catalog numbers, concentrations, and incubation times can be found in Appendix A.

### 2.7. Tumor Histopathology

#### 2.7.1. Tissue Processing and Sectioning

All NBF-fixed tissues were embedded in paraffin wax. First, tissues were processed using a Histocore PEARL (Leica Biosystems) automated tissue processor, using the default overnight protocol. Subsequently, samples were embedded in hot paraffin wax on a heated paraffin embedding module (EG1150 H, Leica Biosystems), before being cooled on a cold plate module (EG1150 C, Leica Biosystems, Nussloch, Germany). Finally, samples were sectioned in 3 μm sections on an RM2125 RTS microtome (Leica Biosystems, Nussloch, Germany) and mounted on glass microscope slides.

#### 2.7.2. Histological Staining

Tumors sections were stained with Mayer’s hematoxylin (Merck, Johannesburg, South Africa) and eosin (Merck, Johannesburg, South Africa) (H&E), utilizing an ST5010 Autostainer XL (Leica Biosystems, Nussloch, Germany) with a modified H&E Program 1 [19]. Finally, sections were mounted with DPX (Merck, Johannesburg, South Africa) mounting media and covered with a glass coverslip.

#### 2.7.3. Pathological Grading and Characterization

Carcinomas were graded and characterized according to the Nottingham combined histologic grading system [20,21] using an Olympus BX41 microscope [high power field (HPF) diameter: 0.55 mm], fitted with an Olympus SC100 camera. Image acquisition was performed using Olympus Stream software (version 2.4, Olympus). Three variables are evaluated with this grading system: the percentage of tubule formation, the extent of nuclear pleomorphism, and the mitotic count in 10 consecutive HPFs. Each variable is assigned a score out of three (1, 2, or 3), for a total score out of nine. Tubule formation was scored as 1 (>75% of tumor area forming glandular/tubular structures), 2 (10% to 75% of tumor area forming glandular/tubular structures), or 3 (<10% of tumor area forming glandular/tubular structures). Nuclear pleomorphism was scored as 1 (nuclei small with little increase in size in comparison with normal breast epithelial cells, regular outlines, uniform nuclear chromatin, little variation in size), 2 (cells larger than normal with open vesicular nuclei, visible nucleoli, and moderate variability in both size and shape), or 3 (vesicular nuclei, often with prominent nucleoli, exhibiting marked variation in size and shape, occasionally with very large and bizarre forms). Mitotic rate was scored as 1 (≤8 per 10 HPFs), 2 (9 to 17 per 10 HPFs), or 3 (≥18 per 10 HPFs), according to the field diameter of the microscope. Lastly, each tumor was assessed for the total percentage of tumor necrosis on a scale ranging from 0 to 100%, before QuPath analysis.

#### 2.7.4. Quantitative Pathology and Bioimage Analysis (QuPath)

Whole slide image acquisition was performed on an ImageXpress^®^ Pico (Molecular Devices, San Jose, CA, USA) with CellReporterExpress^®^ software (version 2.9, Molecular Devices, San Jose, CA, USA), fitted with a 4× objective. Images were subsequently analyzed with QuPath [22] to quantify the total percentage necrosis of each tumor. Here, each tumor’s necrotic region (area) was outlined using a variety of QuPath selection tools and expressed as a percentage of the total tumor area (excluding non-cancerous tissue). Lastly, the total percentage of necrosis for each tumor was calculated.

### 2.8. Data Analyses, Significance, and Presentation

Data were analyzed with GraphPad Prism 9.3.1, wherein outliers were removed using the ROUT (Q = 1%) method, before being assessed for normal/lognormal distribution. Subsequently, the data were analyzed with either an unpaired *t*-test, with or without Welch’s correction, a Mann–Whitney test, or full model 2-way ANOVA correcting for multiple comparisons with Tukey. Statistical significance was set at *p* < 0.05.

## 3. Results

### 3.1. Systemic SAA1/2-Deficiency Does Not Influence Tumor Growth or Host Survival

We investigated the role of acute phase SAA1/2 isoforms in tumor growth, host survival, inflammation, and cellular programs using a breast cancer model. Tumors were induced in genetically wild-type mice (WT) and mice lacking systemic SAA1/2 (SAADKO). Overall, tumor growth showed minimal differences between WT and SAADKO mice (Species main effect, *p* = 0.26) (Figure 1a). 

However, a significant interaction between species and time was observed (*p* = 0.03), as shown in Figure 1a, with significant differences between species on days 32 (*p* = 0.022) and 35 (*p* = 0.043). Survival analysis, as depicted in Figure 1b, using the Cox–Mantel test and Kaplan–Meier cumulative proportional survival test, indicated no significant difference in survival between WT and SAADKO mice (*p* = 0.415).

### 3.2. Systemic SAA1/2-Deficiency Alters Cytokine Expression and Promotes Inflammasome Signaling

Next, we investigated the systemic inflammatory profile of WT and SAADKO mice. In SAADKO control mice (Figure 2a), SAA plasma levels were significantly lower than those of WT control mice (*p* < 0.001), though not entirely ablated. Upon tumor induction, WT mice exhibited a 1.75-fold increase in plasma SAA levels compared to WT control mice (*p* < 0.001). In contrast, tumor induction in SAADKO mice did not affect plasma SAA levels, with both control and tumor-bearing SAADKO mice showing similar levels (*p* = 0.30). As the SAA1/2 genes in SAADKO mice have been replaced with a neomycin resistance gene cassette to generate the double-knockout mice [17], SAA1/2 cannot be produced hepatically in our knock-out mouse model.

Additionally, the inflammatory profile of tumor-bearing SAADKO and WT mice was analyzed (Figure 2b). SAADKO tumor-bearing mice displayed lower concentrations of IL-1β (*p* = 0.030), IL-6 (*p* = 0.003), and IL-10 (*p* = 0.014) compared to WT tumor-bearing mice.

Further analysis revealed that tumors from SAADKO mice showed a reduction in markers associated with inflammasome signaling, including NFκB (*p* = 0.027), caspase 1 (*p* = 0.033), and NLRP3 (*p* = 0.012) (Figure 3). This suggests a suppression of NLRP3 inflammasome activation in SAADKO mice. Representative Western blot images are provided in Appendix A.

### 3.3. Systemic SAA1/2-Deficiency Leads to Lower SAA Levels in Tumors, Reduced Apoptosis Signaling, and Increased DNA Repair

In SAADKO tumors, SAA protein levels were significantly lower (*p* = 0.005) compared to WT tumors (Figure 4). This reduction in SAA protein levels impacted various cellular programs, particularly apoptosis. Specifically, the apoptotic markers cytochrome c (*p* = 0.067), cleaved caspase 9/total caspase 9 (*p* < 0.001), cleaved caspase 3/total caspase 3 (*p* = 0.005), and cleaved PARP/total PARP (*p* < 0.001) were significantly decreased in SAADKO tumors (Figure 4). However, there were no significant changes in cleaved caspase 8/total caspase 8 (*p* = 0.916) and cleaved caspase 7/total caspase 7 (*p* = 0.695) levels (Appendix A).

No significant differences were observed in cell cycle markers, including MCM2 (*p* = 0.107), p53 (*p* = 0.395), and p16INK4a (*p* = 0.350) (Appendix A). Additionally, no significant changes were noted in epithelial-mesenchymal transition (EMT) markers such as E-cadherin (*p* = 0.484), β-catenin (*p* = 0.556), Snail (*p* = 0.095), Laminin 1β (*p* = 0.868), and α-smooth muscle actin (*p* = 0.340) (Appendix A), except for a significant decrease in vimentin (*p* = 0.007).

### 3.4. Systemic SAA1/2-Deficiency Does not Influence Histological Grading

Interestingly, despite lower systemic and tumor SAA levels, histological grading did not improve in SAADKO tumors. On low-power magnification, tumor sections from both SAADKO (6 out of 7) and WT (5 out of 7) mice displayed relatively well-defined, unencapsulated, nodular carcinomas with focal infiltration into surrounding fat and skeletal muscle (Figure 5a,b). The carcinomas were composed of sheets of malignant epithelioid cells without clear glandular differentiation (Figure 5c). On higher magnification, both SAADKO and WT tumors exhibited single-cell necrosis, karyorrhectic debris, and markedly pleomorphic nuclei with vesicular chromatin and large nucleoli (Figure 5d,e). The cytoplasm of the carcinoma cells was voluminous, ranging from eosinophilic to amphophilic. Mitotic activity was high, with more than 100 mitotic figures per 10 high-power fields (HPFs) in both groups, though there was no statistically significant difference in mitotic count between WT and SAADKO tumors (Figure 6). Atypical mitotic figures, including occasional ring forms, were abundant in both groups (Figure 5f–h). All carcinomas were scored as grade 3 according to the Nottingham combined histologic grading system. Additionally, benign ducts were observed in most cases where surrounding native tissue was present (in all SAADKO mice and 5 out of 7 WT mice) (Figure 5i).

### 3.5. Systemic SAA1/2-Deficiency Leads to Lower Tumor Necrosis

WT tumors exhibited more tumor necrosis compared to SAADKO tumors (Figure 6). This was confirmed through both light microscopic assessment (*p* = 0.053) and QuPath analysis (*p* = 0.013), with SAADKO tumors showing significantly less necrosis (Figure 6). Representative sections of WT and SAADKO tumors are presented in Appendix A.

## 4. Discussion

SAA has been associated with various cancer pathologies and its presence in the blood of cancer patients correlates with several clinicopathological features associated with a worse prognosis [6,11,23]. However, the exact function of SAA in cancer pathology has largely remained unclear. We therefore investigated whether acute phase SAA1/2 isoforms could influence tumor growth, host survival, inflammation, and cellular programs in a breast cancer model by inducing tumors in genetically wild-type mice (WT), and mice lacking systemic SAA1/2 (SAADKO).

The results of this study show that the absence of SAA1/2 isoforms in SAADKO mice does not significantly affect overall tumor growth or survival in a breast cancer model, as compared to WT mice (Figure 1). This contrasts with our previous findings in a colitis-associated cancer (CAC) model, where the absence of SAA1/2 led to a significant reduction in tumor formation and a less proliferative tumor phenotype in SAADKO mice [13]. The lack of a similar outcome in the breast cancer model suggests that SAA1/2 may play different roles depending on the cancer type or tissue context.

The significant interaction between species and time observed in tumor growth suggests that the role of SAA1/2 may be more complex and could vary during different stages of tumor development. Despite previous reports linking elevated SAA levels to worse prognosis and disease progression in various cancers, our findings do not show a significant survival advantage or disadvantage in the absence of SAA1/2 in this breast cancer model [24,25,26]. This discrepancy underscores the need for further investigation into the specific mechanisms by which SAA1/2 may influence tumor biology across different cancer types.

Additionally, the results demonstrate that the absence of SAA1/2 in SAADKO mice leads to a significantly altered inflammatory profile compared to WT mice (Figure 2). While tumor induction in WT mice resulted in a significant increase in plasma SAA levels, this was not observed in SAADKO mice, which aligns with the genetic ablation of hepatic SAA1/2 production in this model. The lower concentrations of pro-inflammatory cytokines IL-1β and IL-6 in SAADKO tumor-bearing mice suggest a diminished inflammatory response, which is consistent with findings from our previous CAC study [13]. Interestingly, SAADKO tumor-bearing mice also had reduced levels of IL-10, a cytokine with complex roles in both promoting and resolving inflammation [27,28]. This finding highlights the intricate balance of cytokine signaling in the TME and raises questions about the dual roles of IL-10 in breast cancer pathology [29,30].

Previous studies have indicated a positive feedback loop between SAA and pro-inflammatory cytokines such as IL-1β, suggesting that SAA may play a critical role in amplifying inflammation in the breast TME [31]. The decreased expression of IL-1β in SAADKO mice (Figure 2) correlates with the reduced levels of SAA, further supporting this hypothesis. Additionally, the observed suppression of NLRP3 inflammasome signaling in SAADKO tumors represents a novel finding, as this study is the first to report such an effect in an in vivo breast cancer model (Figure 3). These results suggest that SAA1/2 may be involved in the activation of the NLRP3 inflammasome in breast cancer, contributing to the inflammatory milieu that supports tumor progression [32]. However, given the limited evidence on NLRP3 inflammasome activation in breast cancer, further studies are needed to fully elucidate its role and potential as a therapeutic target in this context.

The findings from this study reveal that the reduction of SAA protein levels in SAADKO tumors is associated with decreased expression of several apoptotic markers, particularly cytochrome c, cleaved caspase 9, cleaved caspase 3, and cleaved PARP (Figure 4). These results are consistent with previous studies, such as those by Kho et al. [33], where SAA1/2/3 expression was found to be linked to apoptosis in the mouse mammary epithelial cell line, HC11 [34,35]. In their study, SAA1/2/3 overexpression suppressed cell growth and activated caspase 8/3/7 under nutrient-starved conditions, which mimic the TME. Similarly, our results suggest that the decreased levels of SAA in SAADKO mice may contribute to reduced apoptosis in tumors. The unchanged levels of cell cycle markers in this study (Appendix A), in contrast to the decreased proliferation markers observed in the CAC model [13], highlight the potential context-dependent role of SAA1/2 in tumor biology. The reduction in the EMT marker vimentin aligns with previous findings that SAA1 knockdown affects vimentin expression [36], suggesting a possible link between SAA and EMT regulation in breast cancer.

Furthermore, the decreased expression of inflammasome-related markers such as NLRP3, along with the observed reduction in IL-1β, suggests a potential interplay between SAA1/2 and inflammasome signaling in regulating apoptosis within the TME. Previous research has shown that genetic ablation of NLRP3 can restore caspase-7 activity in breast cancer cells, indicating a possible connection between inflammasome activation and apoptosis modulation in cancer [37]. Additionally, IL-1 has been shown to rescue tumor cells from apoptosis, further implicating IL-1β as a mediator in this process. However, the possibility of IL-1β-independent mechanisms influencing apoptosis in the absence of SAA1/2 cannot be excluded and warrants further investigation to elucidate the detailed molecular pathways involved.

The results also indicate that lower systemic and tumor SAA levels in SAADKO mice did not lead to improved histological grading (Figure 5), a finding that is at odds with previous studies in human breast tumors where higher SAA levels were associated with worse histological grades [11]. This discrepancy may suggest that the role of SAA in tumor grading and progression could be more complex and context-dependent than previously understood. This finding also contrasts with observations in the CAC model [13], where SAADKO mice exhibited fewer colitis-associated histological abnormalities, pointing to possible differences in SAA’s role across different cancer types and tissues. Additionally, tumors from both SAADKO and WT mice shared similar characteristics, including well-defined nodular carcinomas with focal infiltration into surrounding tissues, and a high degree of cellular pleomorphism and mitotic activity. The lack of significant differences in mitotic counts and the presence of grade 3 carcinomas in both groups suggest that the absence of SAA1/2 does not markedly alter the aggressive histopathological features of breast tumors in this model. Interestingly, the presence of abundant atypical mitotic figures and solid growth patterns, coupled with a lack of glandular differentiation, reinforces the aggressive nature of these tumors, irrespective of SAA1/2 status. The observed similarity in histological features between SAADKO and WT tumors implies that SAA1/2 may not play a pivotal role in modulating the histopathological grade or structure of breast tumors, contrary to its effects observed in other cancer models [11]. Further research is warranted to explore the mechanisms by which SAA1/2 influences tumor biology, particularly in different tissue contexts, and to determine whether other factors might compensate for the absence of SAA1/2 in maintaining the aggressive phenotype of breast tumors.

Lastly, the observation that WT tumors had more necrosis than SAADKO tumors suggests a possible link between SAA and tumor necrosis (Figure 6). This is the first study to directly associate SAA with tumor necrosis, although SAA has been previously correlated with increased HIF1α expression, which is often associated with tumor necrosis [6]. The reduced necrotic burden in SAADKO tumors is consistent with the observed reduction in NLRP3 inflammasome activation and overall inflammation, aligning with a decreased propensity for necrosis. The involvement of the inflammasome in necrotic cell death may provide an explanation for these findings. Activation of the inflammasome can lead to the cleavage of Gasdermin D (GSDMD), creating N-terminal fragments that form pores in the plasma membrane, a key step in pyroptosis and necrotic cell death [38]. The reduced necrosis in SAADKO tumors may therefore be linked to decreased inflammasome activation, resulting in fewer necrotic events. This highlights a potential mechanistic role of SAA in promoting tumor necrosis through inflammasome activation, which could be an important area for future research to better understand the interplay between inflammation, necrosis, and tumor progression.

## 5. Conclusions

Past research has linked SAA proteins to advanced tumor stages, lymphovascular invasion, lymph node metastasis, and increased HIF1α expression [12]. However, the exact impact of elevated SAA levels in cancer remains unclear. Unlike in a previous CAC model where SAA knockout affected tumor growth and inflammation significantly [13], SAA did not influence these parameters in our TNBC model.

Inflammation’s impact on tumors depends on the context and balance of inflammatory signals within the TME. In our breast cancer model, factors like immune cell infiltration and cytokine milieu may inhibit the pro-tumorigenic effects of NLRP3 inflammasome activation. Tumor cells may also adapt by developing resistance to inflammatory signals or exploiting them for survival and immune evasion, resulting in inflammasome activation without increased tumor growth.

The differential outcomes in the two models highlight the tissue-specific roles of SAA proteins in cancer. In the CAC model, SAA promotes local inflammation and tumor progression, while in the TNBC model, it primarily modulates systemic inflammation without significantly affecting tumor growth. This suggests that SAA’s role may be more about modulating the TME rather than directly affecting tumor cell proliferation and survival.

Nonetheless, developing therapeutic strategies targeting SAA expression or its downstream effects could potentially mitigate tumor progression in breast cancer. This approach could improve patient outcomes and enhance existing treatments by addressing the inflammatory component that exacerbates tumor growth and resistance to therapy, making SAA a promising target for novel breast cancer therapies.

## Figures and Tables

**Figure 1 biology-13-00654-f001:**
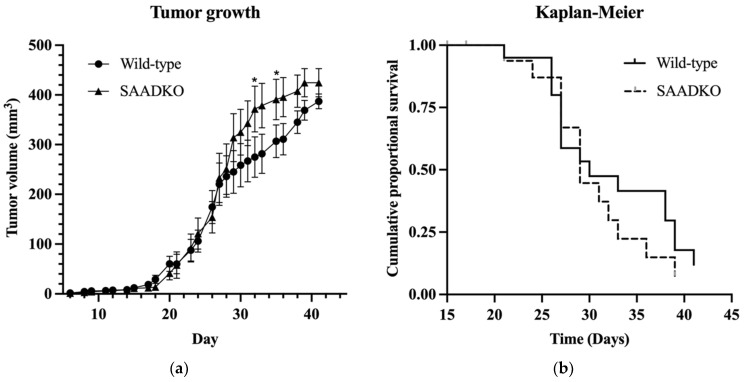
**Tumor growth in wild-type and SAADKO mice and their survival over time.** (**a**) Tumor growth over time for wild-type and SAADKO mice. Data represent the least-squares mean ± SEM (n = 20). (**b**) Kaplan–Meier plot of cumulative proportional survival over time (n = 20). * *p* < 0.05

**Figure 2 biology-13-00654-f002:**
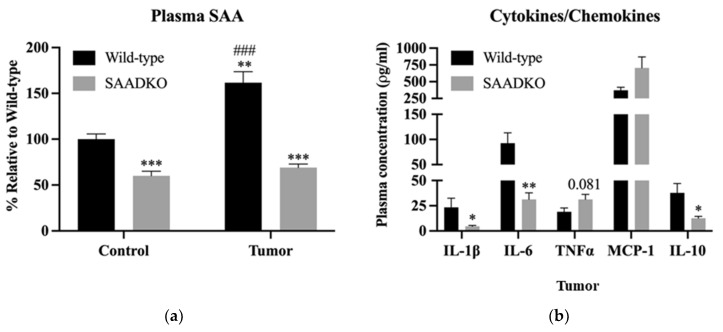
**SAADKO mice have lower systemic SAA levels, which influenced pro- and anti-inflammatory signaling.** (**a**) Plasma SAA was analyzed with a full model 2-way ANOVA correcting for multiple comparisons with Tukey. Data show the mean ± SEM (n = 6). ** *p* < 0.01 control vs. tumor, *** *p* < 0.001 control vs. tumor, ### *p* < 0.001 Wild-type Control vs. Wild-type Tumor. Representative Western blot images for plasma SAA can be found in Appendix A. (**b**) IL-6 was analyzed with an uncorrected unpaired *t*-test. MCP-1 was analyzed with an unpaired *t*-test with Welch’s correction. IL-1β, IL-10, and TNFα were analyzed with a Mann–Witney test. Data show the mean ± SEM (n = 6). * *p* < 0.05 WT vs. SAADKO, ** *p* < 0.01 WT vs. SAADKO.

**Figure 3 biology-13-00654-f003:**
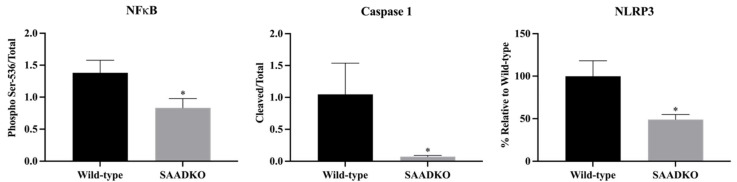
**SAADKO tumors displayed decreased inflammasome activation.** All data were analyzed with an uncorrected unpaired *t*-test. Data show the mean ± SEM (n = 6). * *p* < 0.05 WT vs. SAADKO. Representative Western blot images for NFκB, caspase 1, and NLRP3 can be found in Appendix A.

**Figure 4 biology-13-00654-f004:**
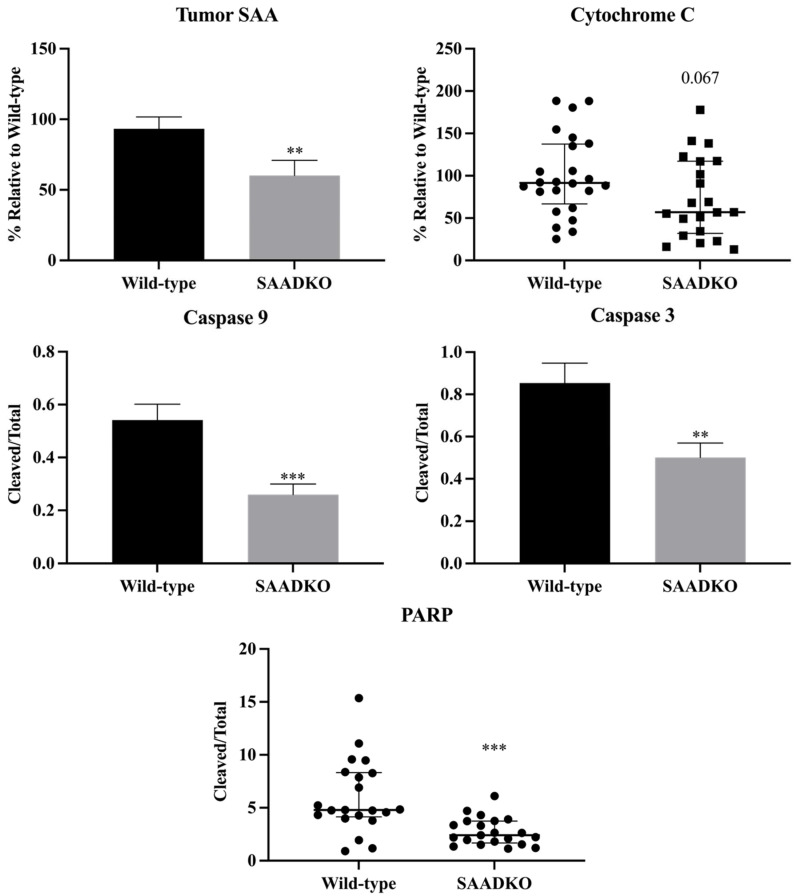
**SAADKO tumors showed lower SAA levels, have reduced apoptosis signaling, and increased DNA repair.** Caspase 3 and caspase 9 were analyzed with an uncorrected unpaired *t*-test. Tumor SAA was analyzed with an unpaired *t*-test with Welch’s correction. Cytochrome c and PARP were analyzed with a Mann–Witney test. Tumor SAA, and Caspase 3/9 data show the mean ± SEM, while cytochrome c and PARP data show the median with IQR (n = 7). ** *p* < 0.01 WT vs. SAADKO, *** *p* < 0.001 WT vs. SAADKO. Representative Western blot images can be found in Appendix A.

**Figure 5 biology-13-00654-f005:**
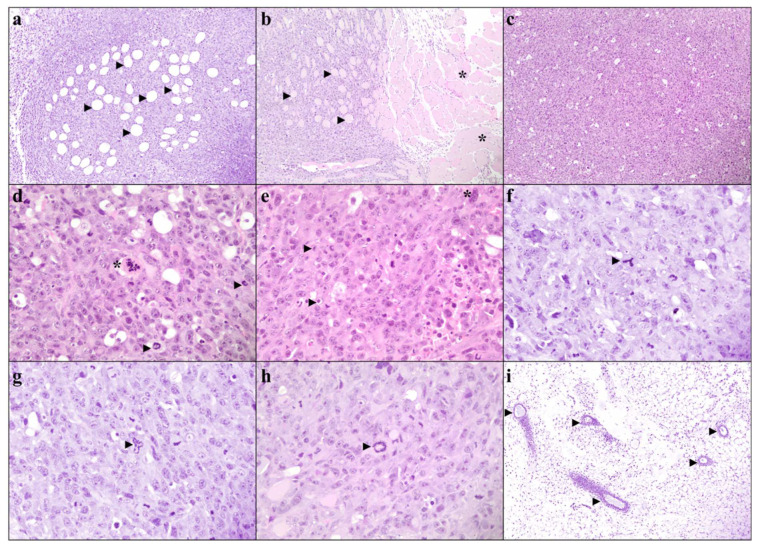
**Tumor pathology of the EO771 syngeneic breast cancer cells in WT and SAADKO mice.** (**a**) Carcinoma infiltrating adipose tissue. Arrowheads show individual adipocytes (WT T60, H&E, 100×). (**b**) Carcinoma infiltrating skeletal muscle. Individual skeletal muscle fibers (arrowheads) can be seen surrounded by sheets of carcinoma cells, while uninvolved skeletal muscle bundles (asterisks) are visible to the right (WT T61, H&E, 100×). (**c**) Diffuse sheets of carcinoma cells without glandular differentiation or other specific architectural features (WT T50, H&E, 100×). (**d**) A single necrotic cell showing cytoplasmic hypereosinophilia and karyorrhexis of the nucleus (asterisk). Scattered mitotic figures (arrowheads) are appreciated in the background (WT T50, H&E, 400×). (**e**) Large eosinophilic nuclei (arrowheads) and multi-nucleic cells (asterisk) were readily present in carcinomas (WT T61, H&E, 400×). (**f**) Tripolar (arrowhead) atypical mitotic figure (WT T60, H&E, 400×). (**g**) Bizarre mitotic figure with multiple mitotic spindles (WT T60, H&E, 400×). (**h**) An atypical ring (arrowhead) mitotic figure (WT T61, H&E, 400×). (**i**) Benign breast ducts (arrowheads) within uninvolved, peritumoral adipose tissue (SAADKO T30, H&E, 100×).

**Figure 6 biology-13-00654-f006:**
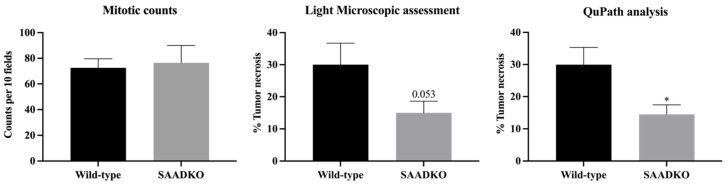
**SAADKO tumors have less tumor necrosis.** All data were analyzed with an uncorrected unpaired *t*-test. Data show the mean ± SEM (n = 7). * *p* < 0.05 WT vs. SAADKO.

## Data Availability

The datasets generated are available in the Mendeley Data Repository at https://data.mendeley.com/datasets/srt3cdxznz/1 (accessed on 20 July 2024).

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
