# Peer review of "Inflammation and Tumor Progression: The Differential Impact of SAA in Breast Cancer Models"

_biology, 2024, doi:10.3390/biology13090654_

Round 1

Reviewer 1 Report

Comments and Suggestions for Authors

In this manuscript, the authors examine the contribution of Serum Amyloid A (SAA) in tumor molecular biology. Below are my specific comments/suggestions.

1.      From Figure 2 results, please provide justifications why several different statistical methods were used for different cytokines/chemokines (as opposed to using one method consistently for all the examined cytokines/chemokines).

2.      One major drawback of this work is that only one tumor model (from EO771 cancer cell line) was evaluated. As breast cancer is heterogenous in clinical settings, adding additional tumor models would further confirm the translatability of the manuscript’s finding. However, this would involve in vivo experimentation and may thus require significant time/resources to conduct, so I leave it to the discretion of the authors and the editors to decide whether completion of such studies is required prior to accepting the work for publication.

3.      Some of the acronyms are missing their definitions during their first usage including: CARD domain (line 80), PAMPs/DAMPs (line 88).

4.      Incomplete sentence (line 86)

5.      To avoid confusion from the reader, please consider indicating within Figure 2, panel right that these cytokines/chemokines measurements are from tumor-bearing mice (not control mice).

Comments on the Quality of English Language

In general, the manuscript is well-written.

Reviewer 2 Report

Comments and Suggestions for Authors

I appreciate that the authors have done very novel and extensive work to investigate the role of systemic SAA1 and SAA2 (SAA1/2) in a triple-negative breast cancer mouse model. This study builds on previous research showing that SAA proteins are involved in inflammation and various diseases, including colitis and tumorigenesis. By comparing wild-type mice with mice lacking SAA1/2 (SAADKO), the authors found that WT mice with tumors had higher SAA levels in their blood, while those lacking SAA1/2 had lower levels and less inflammation. Additionally, tumors from mice lacking SAA1/2 showed reduced cell death and fewer signs of severe tissue damage. These findings suggest that SAA1 and SAA2 increase inflammation and cell death in breast tumors but do not significantly affect tumor growth. The novelty of this work is rational and may contribute valuable insights into the role of SAA1/2 in breast cancer pathology.

However, I have a few doubts and questions that I would like the authors to address, please:

  1. Could the authors please discuss the genotyping results and the gel pictures used? It appears that a significant amount of genomic DNA was utilized for PCR, which might lead to non-specific amplifications.
  2. What do the authors think about the possibility of functional redundancy being taken up by SAA3/4 in the context of the SAA1/2 double knockdown? The probability of such genetic redundancy was not discussed in the manuscript.
  3. In Appendix D, I couldn't find any loading controls, such as GAPDH or beta-actin, to compare the protein amounts. I would request the authors to please clarify and rewrite the figure legends, as some of them are unclear and do not adequately explain the figures.
  4. Could the authors please discuss the supplementary materials included in the appendices? Some of the supplementary materials are not mentioned or discussed in the main paper, and I would appreciate an explanation for their inclusion.
  5. The authors claim that NLRP3 is regulated by SAA1/2. To substantiate this claim, I believe the authors should perform pull-down experiments to show that NLRP3 is directly regulated by SAA1/2. Could the authors please address this?
  6. Why does the SAA1/2 double knockout not show any differences in the H&E (Histology) staining, despite showing effects on IL-1β, IL-6, and TNFα? While it is understood that function can be context-dependent, there is a change in the wild-type tumor after induction (Figure 1). Do the authors believe these changes occur without any morphological differences?
  7. Why have the authors not performed Ki67 staining? If there were no differences observed, it would be helpful to confirm this with Ki67 staining, especially since apoptosis is observed. The previous paper by the authors included Ki67 staining, so could they please explain why it was omitted in this study?

Thank you for considering these questions and clarifications. I believe addressing them will enhance the understanding and impact of your valuable research.

Regards.

Reviewer 3 Report

Comments and Suggestions for Authors

In this manuscript, that systemic SAA1/2 stimulates the activation of the NLRP3 inflammasome in breast tumors was demonstrated. The method of the article is rigorousand the overall context of the article is smooth and clear. However, this article has also some issues that should be addressed for further consideration:

1. The article is about SAA, but the keywords do not include SAA.

2. Are there references to the various methods used in the article? If so, add the corresponding references.

3. There should be spaces between symbols and letters in the article(Acute-phase proteins;Breast cancer).

4. Check the format of usingTrypLE™️ Express.

5. Abbreviations in the article should be consistent(10 seconds and boiled for 5 min).

6. “Western blot” should be capitalized.

7. Redundant information in references should be removed(https://doi.org/10.3892/ol.2012.584).
